# EXPLOITING CERTIFIED DEFENCES TO ATTACK RANDOMISED SMOOTHING

## ABSTRACT

Certified guarantees of adversarial robustness play an important role in providing assurances regarding a models output, irrespective of the behaviour of an attacker. However, while the development of such guarantees has drawn upon an improved understanding of attacker behaviour, so too can certified guarantees be exploited in order to generate more efficient adversarial attacks. Within this work, we explore this heretofore undiscovered additional attack surface, while also considering how previously discovered attacks could be applied to models defended by randomised smoothing. In all bar one experiment our approach generates smaller adversarial perturbations for more than $70\%$ of tested samples, reducing the average magnitude of the adversarial perturbation by $13\%$.

## 1 INTRODUCTION

The observation that neural networks' exhibit particular sensitivity to adversarial behaviours has motivated numerous recent works. That such networks are frequently deployed within contexts for which incentives for adversarial behaviour exist makes guarding against such interventions to be of paramount importance.

Our understanding of this problem space has been heavily driven by the development of new perspectives on potential attack vectors, stemming the original works in adversarial examples through to data poisoning, backdoor attacks, model stealing, transfer attacks and more. While uncovering such attack vectors has the potential to compromise deployed models, there is prima facie evidence that any security provided by a lack of knowledge is illusory. As such, it is clear that understanding new attack vectors has the potential to produce stronger defences, due to the implicit coupling between attacks and their defences.

In contrast to this implicit coupling, recent works have focused upon the construction of *certified guarantees of adversarial robustness*, which verify that a models output will be unchanged over *all* adversarial behaviours (subject to a set of loose constraints). The calculations of such guarantees are performed in parallel to inference, and are achieved by way of modifications to the mechanisms of the model. One common backbone for such techniques is randomised smoothing, which can be performed without requiring any modifications to the core training loop beyond a pre-processing and post-processing step.

However, while models incorporating certified guarantees should be more adversarially robust than their counterparts, within this work we consider how such models can still be attacked in a fashion that yields small adversarial perturbations. Such a consideration is important, as we also identify a heretofore undiscovered new attack vector, which exploits the very nature of the certified guarantee to identify smaller adversarial perturbations than any other tested approach. To explore the nature of this novel attack surface, this work makes the following contributions:

- Demonstrating how general attacks can be constructed against models defended by randomised smoothing. This is achievable by exploiting the Gumbel-Softmax within either white-box or surrogate modelling attacks to render the models differentiable. In doing so, we are able to exploit the fact that randomised smoothing inherently smooths the underlying gradients, making them easier to attack.
- Introducing a new attack that exploits certification guarantees in *both the original and malicious classes* by dynamically optimising the step-size. Correct class predictions exploit

the certification to *eliminate* parts of the search space, incorrect class predictions are used to *focus* the search process. In doing so we are able to identify smaller adversarial perturbations than the best tested alternative in over $70\%$ of all tested samples, resulting in a more than $13\%$ reduction in the median perturbation.

- Assessing the performance of certified guarantees of adversarial robustness, through comparison to the magnitude of best-identified adversarial perturbations in $L_2$ norm space.

## 2   RELATED WORK

It is well known that carefully crafted perturbations can change the output of learned models without requiring change in the semantic properties of the input sample (Biggio et al., 2013). Such perturbed samples are known as *adversarial examples*. Most common learned models, including neural networks (Szegedy et al., 2013) consistently misclassify adversarial examples and furthermore output highly confident but incorrect predictions. One significant driver of this behaviour appears to be the piecewise-linear interactions within neural networks (Goodfellow et al., 2014).

While many attacks exist, we focus on key representative approaches. Each of these are white-box, untargeted attacks that use gradient-based optimisation to construct adversarial perturbations. Of these, the Iterative Fast Gradient Method (Dong et al., 2018) variant of Projected Gradient Descent (PGD) (Carlini & Wagner, 2017) allows adversarial examples to be iteratively constructed by way of

$$\mathbf{x}_{k+1} = P\left(\mathbf{x}_k - \epsilon\left(\frac{\nabla_{\mathbf{x}} J(\theta, \mathbf{x}, y)}{\|\nabla_{\mathbf{x}} J(\theta, \mathbf{x}, y)\|_2}\right)\right) \quad . \tag{1}$$

This process exploits gradients of the loss $J(\theta, \mathbf{x}, y)$ to construct steps, subject to a step-size weighting parameter $\epsilon$, and a projection operator $P$ that ensures that $\mathbf{x}_{k+1}$ is restricted to the feasible input space, is typically $[0,1]^d$ for a $d$-dimensional input space. Many PGD extensions exist, including momentum-based variants (Dong et al., 2018) and AutoAttack (Croce & Hein, 2020).

The latter of these alternate approaches has been shown to be highly effective in identifying adversarial examples, and as such we also tested performance against AutoAttack within this work. In contrast to PGD, which sets a fixed step-size $\epsilon$, AutoAttack algorithmically specifies the step-size at each stage of its iterative process. Instead AutoAttack attempts to converge upon adversarial examples with a pre-specified $L_2$ norm perturbation magnitude, which obtusely is also labelled as $\epsilon$. This is inherently problematic within problem domains for which minimising the perturbation magnitude is important, the requirement to pre-specify the perturbation magnitude is an inherently limiting factor. Our preliminary investigations have suggested that the only way to minimise the perturbation magnitude is to perform a greedy search over a range of possible pre-specified magnitudes.

Carlini & Wagner (C-W) constructs adversarial perturbations by way of the minimisation problem

$$\min_{\mathbf{x}'} \|\mathbf{x}' - \mathbf{x}\|_2^2 + \max\left\{\max\{f_\theta(\mathbf{x}')_j : j \neq i\} - f_\theta(\mathbf{x}')_i, -\kappa\right\} \tag{2}$$

in terms of the trained model $f_\theta(\mathbf{x})$ (with weights $\theta$). The latter term of Equation 2 compares the logit value of the target class $i$ with that of the next most likely class, subject to the parameter $\kappa$. This criteria is then solved in the fashion of Equation 1, using gradients from Equation 2.

Another popular attack is DeepFool, which is an untargetted $L_2$-norm attack (Moosavi-Dezfooli et al., 2016) that interchanges between attacking a linearised variant of the model and updating the linearisation based upon gradient steps. This linearisation allows for automatic step-size control across the iterative process.

## 3   ATTACKS AGAINST CERTIFIED DEFENCES

Randomised smoothing is a common approach for constructing certified guarantees by constructing expected class outputs through Monte Carlo sampling across samples perturbed by randomised noise. While this is a test-time process, the sensitivity of the model to perturbations may be decreased by performing adversarial training against single draws from the noise distribution. The magnitude of the guarantee of predictive invariance for perturbations up to some magnitude $r$ in a specified $L_p$-norm space was first shown by way of differential privacy (Lecuyer et al., 2019; Dwork et al., 2006). More

recent works has utilised Rényi divergence (Li et al., 2019) and worst-case behaviours parametrisation (Cohen et al., 2019; Salman et al., 2019) to provide guarantees. We place particular focus upon the latter of these approaches, which introduced the lower bound guarantee of class invariance

$$r = \frac{\sigma}{2} \left( \Phi^{-1}(E_0) - \Phi^{-1}(E_1) \right) \quad , \tag{3}$$

subject to the normal CDF $\Phi$, in terms of the two largest class expectations $E_0$ and $E_1$, which are calculated in terms of a one-hot encoding of the model's output in response to noise

$$\tilde{\mathbf{X}} \sim \mathcal{N}(\mathbf{x}, \sigma^2 \mathbf{I}) \qquad \tilde{\mathbf{Z}} = f_{\boldsymbol{\theta}}(\tilde{\mathbf{X}}) \qquad \tilde{Y}_j = \begin{cases} 1, & \tilde{Z}_j > \max_{k \in \mathcal{K} \setminus j} \tilde{Z}_k \\ 0, & \text{otherwise.} \end{cases} \qquad \mathbb{E}_{\mathcal{D}}[\tilde{\mathbf{Y}}] = \frac{1}{n} \sum_{i=1}^{n} \tilde{\mathbf{y}}_i \quad . \tag{4}$$

We are interested in estimating $\mathbb{E}[\tilde{\mathbf{Y}}]$ as our ideal smoothed predictions. By taking a sample $\mathcal{D}$ of $n$ i.i.d. draws $\tilde{\mathbf{x}}_i, \tilde{\mathbf{z}}_i, \tilde{\mathbf{y}}_i, \tilde{\mathbf{y}}_i'$ of these random variables, we may form unbiased estimates of these ideal expectations by expectations with respect to the empirical distribution on $\mathcal{D}$.

In the absence of randomised smoothing, the objective of an attack is simple: to change the predicted class of the model. However, under randomised smoothing that the output is expectations (with an associated confidence interval) introduces an important question regarding the nature of an attack within this context. If an idealised output has a no overlap between the confidence interval of the class with the highest expectation and those from all other classes, is it enough for an attack to simply introduce an overlap between the confidence intervals associated with the largest two classes? While such a definition has utility, the overlap between predicted classes would be a simple trigger for external validation, decreasing the likelihood of success. This in turn suggests that an adversarial perturbation should instead be defined as one for which the associated class has an associated confidence interval that is strictly larger than that of the label class. Such a definition naturally aligns with that of certified defences of adversarial robustness.

## 3.1 THREAT MODEL

To formally define such an attack, consider a model $f_{\theta}(\mathbf{x}; \sigma, N)$ which acts upon inputs $\mathbf{x} \in [0,1]^d$ and learned parameters $\theta$, which produces a prediction of class $i$ if the estimated expectation—based upon adding $N$ draws of $\mathcal{N}(0, \sigma^2)$ Gaussian distributed noise to the input sample—satisfies

$$\widecheck{\mathbb{E}}_i[f_{\theta}(\mathbf{x})] > \widehat{\mathbb{E}}_k[f_{\theta}(\mathbf{x})] \qquad \forall k \in \mathcal{K} \setminus i \quad , \tag{5}$$

where $\widecheck{E}_i$ and $\widehat{E}_k$ respectively represent the lower and upper confidence bounds on class expectations of classes $i$ and $k$ to some confidence level $\alpha$ (as calculated by way of the Goodman et al. (Goodman, 1965) confidence interval), and $\mathcal{K}$ is the set of possible output classes. An attack on such a classification is then a sample $\mathbf{x}' \in [0,1]^d$ for which

$$\widecheck{\mathbb{E}}_j[f_{\theta}(\mathbf{x}')] > \widehat{\mathbb{E}}_k[f_{\theta}(\mathbf{x}')]$$
$$\text{where } i = \arg\max_{m \in \mathcal{K}} \mathbb{E}_m\left[f_{\theta}(\mathbf{x})\right], \exists j \in \mathcal{K} \setminus i, \ \forall k \in \mathcal{K} \setminus j \quad . \tag{6}$$

The very nature of randomised smoothing makes such a definition particularly amenable to gradient-based attacks. Randomised smoothing can be thought of as a Gaussian blur of the decision space, which both removes many isolated adversarial examples, and decreases the local variance of gradients in this space. This latter property further enhances the performance of iterative gradient based attacks.

The process that we will now describe requires white-box access to model and its parameter space, including the level of added noise $\sigma$. While such a white-box attack framework is limiting, previous work has demonstrated that it may be possible to successfully attack black-box models by way of surrogate models (Papernot et al., 2017), effectively converting black-box models into white-box's suitable for attack. Moreover, as will be discussed in Appendix E.1, the attacker does not require exact knowledge of $\sigma$, with even approximate values still yielding an attack which exhibits improved performance relative to comprable attacks.

## 3.2 ATTACKING THROUGH UNDIFFERENTIABLE LAYERS

That the calculation and sorting of expectations involves undifferentiable $\arg\max$ layers inherently limits the applicability of gradient-based attacks to such a problem space. However, stochastic

gradient estimation techniques have been shown to accurately generate approximate derivatives, even across undifferentiable layers (Fu, 2006; Chen et al., 2019). Within this work we implement gradient-based attacks by replacing the $\arg\max$ layer with the Gumbel Softmax (Jang et al., 2016)

$$y_i = \frac{\exp\left((\log(\pi_i) + g_i)/\tau\right)}{\sum_{j\in\mathcal{K}} \exp\left((\log(\pi_i) + g_i)/\tau\right)} \text{ for all } i \in \mathcal{K} \tag{7}$$

in terms of temperature $\tau$ and i.i.d samples $g_i \sim \text{Gumbel}(0,1)$, which approximates the $\arg\max$ operation in the $\tau \to 0$ limit. As will be discussed in Section 4 these modifications allow for the application of gradient-based attacks like PGD, Carlini-Wagner, DeepFool, and AutoAttack to models defended by randomised smoothing.

### 3.3 ENHANCED CERTIFICATION-AWARE ATTACKS

While making such modifications allows for attacks to be applied to models defended by randomised smoothing, it is also possible to construct new attack frameworks that incentivise the construction of small adversarial perturbations against models defended by randomised smoothing. Focusing upon small adversarial examples is crucial as such examples are likely to decrease the likelihood of the attack being detected (Gilmer et al., 2018). To achieve this, we introduce the attack

$$\min_{\mathbf{x}' \in [0,1]^d} \left| \max_{j \in \mathcal{K}\backslash i} \breve{\mathbb{E}}_j[f_\theta(\mathbf{x}')] - \widehat{\mathbb{E}}_i[f_\theta(\mathbf{x}')] \right| + \lambda \|\mathbf{x}' - \mathbf{x}\| \ . \tag{8}$$

for some class $j \in \mathcal{K}$, if $\arg\max \mathbb{E}[f_\theta(\mathbf{x})] = i$, subject to the Lagrange multiplier $\lambda$. This formalism may appear counter-intuitive, as we are seeking to identify the smallest example $\mathbf{x}'$ such that $\|\mathbf{x}' - \mathbf{x}\|$, which traditionally would lead to the Lagrange multiplier being applied to the first term of Equation 8. Constructing the objective of the minimisation process in such a fashion prioritises finding parts of the parameter space likely to contain an adversarial example, before converging upon solutions that prioritise minimising the distance between $\mathbf{x}'$ and $\mathbf{x}$.

This formalism admits solutions constructed following the iterative process

$$\mathbf{x}'_{k+1} = P\left(\mathbf{x}'_k - s\frac{d}{\|d\|_2}\right) \text{ where} \tag{9}$$

$$d = \nabla_{\mathbf{x}'}\left(\left|\max_{j\in\mathcal{K}\backslash i} \breve{E}_j[f_\theta(\mathbf{x}')] - \widehat{E}_i[f_\theta(\mathbf{x}')]\right| + \lambda\|\mathbf{x}' - \mathbf{x}\|\right)$$

for a given step-size $s$. Similar to PGD, the projection operator $P$ is defined such that $\mathbf{x}'_{k+1} \in [0,1]^d$.

**Step-size control by certified robustness**  Attacking models defended by randomised smoothing presents additional opportunities to improve attack performance. We can exploit the fact that robustness certificates allow us to generate a guarantee of class invariance for perturbations with $L_2$ norm bounded magnitude less than $r$. This then allows the step-size $s$ from $\mathbf{x}_0 \to \mathbf{x}'_1$ to be set such that $s > r$, as we are guaranteed that no adversarial example can exist within this radius. Introducing this change allows for a significant reduction of the potential search space for adversarial examples. This process also applies to any $\mathbf{x}'_k$, producing certifications with an $L_2$ norm magnitude of

$$r_k = \frac{\sigma}{2}\left(\Phi^{-1}\left(\breve{E}_0\left(\mathbf{x}'_k\right)\right) - \Phi^{-1}\left(\widehat{E}_1\left(\mathbf{x}'_k\right)\right)\right) \ , \tag{10}$$

by way of the two largest class expectations $E_0$ and $E_1$. If $E_0$ corresponds to the original predicted class, then setting $s_k > r_k$ *guarantees that steps take* $\mathbf{x}'_{k+1}$ *outside the region of the adversarial guarantee*. However, if an adversarial perturbation has been constructed such that $E_0$ corresponds to a different class, then setting $s_k < r_k$ guarantees that $\mathbf{x}'_{k+1}$ *will also remain an adversarial example*.

While this efficiently traverses the potential search space, it is possible that large steps may move the iterative process towards an early local optimum, bypassing a pathway towards convergence to a preferred optimum. As such, our implementation incorporates a maximum step-size $c$, such that

$$s_k = \min\{m \times r_k, c\} \ , \tag{11}$$

where $m > 1$ if the predicted class at $\mathbf{x}'_k$ matches the predicted class $i$, and $m \leq 1$ if the classes do not match, as is seen on lines 14 and 18 of Algorithm 1. While bounding the step-size in terms of the certified radius enhances the ability to converge upon adversarial examples, enforcing the maximum step-size disincentivises large steps that may lead towards the convergence of a local optima.

---

**Algorithm 1** Certification Aware Attack Algorithm.

---

1: **Input:** data $\mathbf{x}$, level of additive noise $\sigma$, samples $N$, iterations $M$, true-label $i$, cutoff $c$, multiplicative Lagrange multiplier scaling factor $\lambda$, penalty parameters $\Gamma \gg 1$ and $\delta \geq 0$
2: $\mathbf{x}', \mathbf{x}'_s, m = \mathbf{x}, \mathbf{0}, \infty$
3: **for** 1 **to** $M$ **do**
4:    $\breve{E}_0, \widehat{E}_1, R = \text{Model}(\mathbf{x}'; \sigma, N)$ {Detailed in Algorithm 2}
5:    **if** $\arg\max y \neq i$ **then**
6:      **if** $\breve{E}_0 > \widehat{E}_1$ **then**
7:        $d = \nabla_{\mathbf{x}'}\left((\breve{E}_0 - \widehat{E}_1) + \lambda\|\mathbf{x}' - \mathbf{x}\|\right)$
8:        **if** $\|\mathbf{x}' - \mathbf{x}\| < m$ **then**
9:          $m, \mathbf{x}'_s = \|\mathbf{x}' - \mathbf{x}\|, \mathbf{x}'$
10:        **end if**
11:      **else**
12:        $d = \nabla_{\mathbf{x}'}\left(\Gamma(\breve{E}_0 - \widehat{E}_1 - \delta) + \lambda\|\mathbf{x}' - \mathbf{x}\|\right)$ {$\Gamma$ is promotes distinct predictions}
13:      **end if**
14:      $s = \min\{0.99R, c\}$ {As $R$ is *about* $\mathbf{x}'$, this step is guaranteed to retain the class prediction}
15:      $\lambda = L\lambda$
16:    **else if** $\arg\max y = i$ **then**
17:      $d = \nabla_{\mathbf{x}'}(\breve{E}_0 - \widehat{E}_1 + \delta)$
18:      $s = \min(1.05R, c)$ {$s > R$ is necessary to inducing a change in the predicted class}
19:    **end if**
20:    $\mathbf{x}' = P(\mathbf{x}' - s\frac{d}{\|d\|_2})$ {Project upon $[0,1]^d$}
21: **end for**
22: **return** $m, \mathbf{x}'_s$

---

**Algorithmic innovations** This section has demonstrated both *how models defended by randomised smoothing can be attacked* and how exploiting Equation 3 as an *additional attack surface by significantly reducing the adversarial example search space*, in order to attempt to identify *the smallest perturbation that is still an adversarial example*.

Algorithm 1 contains several additional features designed to enhance the convergence upon the smallest identifiable adversarial example, which we highlight now. One of these is the parameter $\delta > 0$ on lines 12 and 17, which promotes identifying distinct adversarial examples, ones for which $\arg\max_{j \in \mathcal{K}} E_j[f_\theta(\mathbf{x}'_k)] \neq i$ and $\max_{j \in \mathcal{K} \setminus i} \breve{E}_j[f_\theta(\mathbf{x}'_k)] > \widehat{E}_i[f_\theta(\mathbf{x}'_k)]$, promoting a small degree of additional separation beyond the introduced bounds.

A final feature designed to promote the identification of distinct adversarial examples can be seen in the introduction of $\Gamma \gg 1$ into line 12. In the case of a non-distinct adversarial example, this parameter incentivises the iterative process to converge upon a distinct example, by changing the balance of weights between the terms relating to class difference and sample distances.

Additional algorithmic nuance is found on on lines 7 and 12 of Algorithm 1. Initially the Lagrange multipliers introduce a counterproductive bias towards minimising $\|\mathbf{x}' - \mathbf{x}\|$, as our step-size control ensures that are efficiently moving towards nearby adversarial examples. After such an example has been found, Line 14 ensures that the predicted class will not change, with the Lagrange multiplier of lines 7 and 12 then incentivising minimising $\|\mathbf{x}' - \mathbf{x}\|$.

Code demonstrating both our process, and our tested comparisons can be found at *anonymised link*.

## 4 EXPERIMENTS

To evaluate both the performance of our new attack technique and the difference between identified adversarial perturbations and certified guarantees, we now present comprehensive experimental validation against MNIST (LeCun et al., 1998) (GNU v3.0 license), CIFAR-10 (Krizhevsky et al., 2009) (MIT license), and Tiny-Imagenet (Johnson et al., Accessed 2022-01-10) (BSD 3−Clause license), the latter of which is a 200-class variant of Imagenet (Yang et al., 2021) which downsamples

images to $3 \times 60 \times 60$. Each model was trained in PyTorch (Paszke et al., 2019) using a Resnet18 architecture, with experiments considering two distinct levels of $\sigma$. The confidence intervals of the expectations were calculated at a significance level of $\alpha = 0.005$. Based upon Appendix E.3, the cutoff factor $c$ of Equation (11) was set to $0.5$.

For both MNIST and CIFAR-10 the experimentation employed a single NVIDIA P100 GPU core with 12 GB of GPU RAM, with expectations estimated over 1500 samples. Over the course of 50 epochs of training, each sample was perturbed with a single perturbation drawn from $\mathcal{N}(0, \sigma^2)$ and added prior to normalisation. Training then utilised a batch size of 128, with losses assessed against the Cross Entropy loss. Parameter optimisation was performed with Adam (Kingma & Ba, 2014), with the learning rate set as $0.001$. Tiny-Imagenet training and evaluation utilised 4 P100 GPU's, utilising a total of 48 GB of GPU RAM to estimate expectations based upon 1000 samples. Training occurred using SGD over 80 epochs, with a starting learning rate of $0.1$, decreasing by a factor of 10 after 30 and 60 epochs, and momentum set to $0.9$.

To assess the relative performance between our newly constructed adversarial perturbations and the corresponding certified guarantees (as provided by Cohen et al., 2019) Equation (3) provides a lower bound on the distance between an adversarial example and the sample point. To aide this comparison, we introduce the concept of the *attack proportion*, which represents the proportion of correctly predicted samples that have an identified attack below a given $L_2$-norm radius. The lower bound attack radius—as provided by the certified guarantee—gives an upper bound on the achievable attack proportion for any given radius. Our new Certification Aware Attack framework is also compared against PGD, Carlini-Wagner and DeepFool based upon 100 attack iterations, with metrics collected after the first successful attack and at the end of the iterative process. The nature of AutoAttack requires the attack radii to be pre-specified, which we set at $100\%$ increase over the radius identified by Cohen et al. (2019). Further details of the attack hyperparameters can be found in Appendix C.

**Performance against other attacks** Across the full set of tested experiments, the aggregate measures of Figure 2 and Table 1 demonstrate that our Certification Aware Attack almost uniformly identifies smaller adversarial perturbations than any other technique, resulting in a $24\%$ reduction in the median perturbation radii for Tiny-Imagenet at $\sigma = 1.0$, relative to the next best performing technique in PGD. Across the full suite of tested experiments, on average our technique yields a $13.6\%$ reduction in the median perturbation radii. Of the remaining techniques, both DeepFool and Carlini-Wagner exhibit median radii perturbation radii that is an order of magnitude larger than both our technique and PGD, although in the case of DeepFool these examples are identified significantly faster than any other tested approach. Even setting the attack radii to be significantly larger than the median radii identified by our approach and PGD, AutoAttack was able to identify significantly fewer adversarial examples than our approach.

It is important to note that while increasing $\sigma$ should improve the performance of all gradient-based attacks due to the smoothing influence of noise upon the label space. However it is clear that our attack is particularly well suited to this change, with a distinct improvement relative to PGD within Figure 2.

Neither PGD nor DeepFool were able to converge upon smaller adversarial perturbations with additional iterations, in contrast to the design of both our approach and Carlini-Wagner. Table 1 demonstrates that our Certification Aware Attacks can reduce the magnitude of the identified adversarial perturbation $15\%$, at the cost of a $2 - 3$-fold increase in computational time. While continuing to iterate with Carlini-Wagner yields more significant improvements even after the 100 iterative steps, the attacks are still an order of magnitude larger than our approach, and take more than twice as long.

Of the experimental set, the results observed for MNIST deserve additional consideration, due to their surprising nature. While MNIST has a limited number of classes, single-channel low-resolution images, and a perceived lack of semantic complexity across the class set, it proved the most difficult example to attack, with only Carlini-Wagner being able to attack more than $90\%$ of samples. That these issues are more pronounced for $\sigma = 0.5$ intuitively suggests that this may be due to each attack using step-sizes that are too large for the smaller, single-channel input space of MNIST. Increasing $\sigma$ adds additional smoothing to the label space and pushes adversarial examples further out. However, both our Certification Aware Attack and DeepFool do not rely upon fixed step-sizes at all, but rather construct their step-sizes by drawing upon observations of the state. This suggests that the relatively

Table 1: Performance metrics for MNIST (M), Cifar-10 (C), and Tiny-Imagenet (TI) for varying $\sigma$. 'Success' and 'Best' are the proportion samples for which each attack was success, and outperformed all others. $r_{50}$ and %-Cohen are the median attack and the size relative to the guarantee of Cohen.

| Categorisation | | | Smallest Attack | | | | First Attack | |
| Data | Attack | Success | Best | $r_{50}$ | %-Cohen | Time (s) | Ratio($r_{50}$) | Ratio(Time) |
| --- | --- | --- | --- | --- | --- | --- | --- | --- |
| M-0.5 | Ours | 75% | 71% | 1.95 | 67% | 5.49 | 1.10 | 0.23 |
| | PGD | 65% | 6% | 1.95 | 69% | 8.95 | 1.00 | 0.34 |
| | C-W | 94% | 25% | 8.19 | 612% | 5.29 | 1.63 | 0.02 |
| | Auto | 31% | 2% | 1.95 | 100% | 174.87 | 1.00 | 1.00 |
| | D.Fool | 5% | 1% | 9.20 | 2107% | 0.66 | 1.00 | 1.00 |
| M-1.0 | Ours | 100% | 89% | 2.41 | 66% | 2.87 | 1.09 | 0.43 |
| | PGD | 98% | 4% | 2.75 | 88% | 8.92 | 1.00 | 0.43 |
| | C-W | 94% | 0% | 8.83 | 505% | 4.49 | 1.51 | 0.03 |
| | Auto | 90% | 6% | 3.05 | 100% | 174.08 | 1.00 | 1.00 |
| | D.Fool | 47% | 0% | 12.80 | 1409% | 0.65 | 1.00 | 1.00 |
| C-0.5 | Ours | 96% | 83% | 0.85 | 69% | 3.34 | 1.23 | 0.19 |
| | PGD | 95% | 3% | 0.97 | 88% | 9.19 | 1.00 | 0.14 |
| | C-W | 95% | 2% | 6.65 | 1104% | 4.58 | 2.07 | 0.03 |
| | Auto | 85% | 11% | 1.27 | 100% | 179.28 | 1.00 | 1.00 |
| | D.Fool | 90% | 0% | 2.39 | 512% | 0.67 | 1.00 | 1.00 |
| C-1.0 | Ours | 100% | 85% | 1.24 | 79% | 2.38 | 1.13 | 0.38 |
| | PGD | 100% | 1% | 1.55 | 114% | 9.17 | 1.00 | 0.22 |
| | C-W | 95% | 0% | 6.76 | 775% | 4.43 | 2.01 | 0.03 |
| | Auto | 74% | 12% | 1.88 | 100% | 178.91 | 1.00 | 1.00 |
| | D.Fool | 98% | 1% | 3.05 | 462% | 0.67 | 1.00 | 1.00 |
| T-I-0.5 | Ours | 94% | 77% | 0.97 | 89% | 6.62 | 1.18 | 0.23 |
| | PGD | 93% | 1% | 1.15 | 113% | 18.80 | 1.00 | 0.17 |
| | C-W | 100% | 6% | 11.79 | 1981% | 13.76 | 2.45 | 0.02 |
| | Auto | 67% | 15% | 1.38 | 100% | 360.88 | 1.00 | 1.00 |
| | D.Fool | 79% | 1% | 2.39 | 532% | 1.36 | 1.00 | 1.00 |
| T-I-1.0 | Ours | 95% | 84% | 1.37 | 107% | 4.70 | 1.13 | 0.44 |
| | PGD | 94% | 1% | 1.80 | 161% | 19.51 | 1.00 | 0.26 |
| | C-W | 100% | 5% | 10.73 | 1373% | 14.38 | 2.73 | 0.02 |
| | Auto | 54% | 11% | 2.29 | 100% | 372.23 | 1.00 | 1.00 |
| | D.Fool | 89% | 0% | 3.44 | 577% | 2.72 | 1.00 | 1.00 |

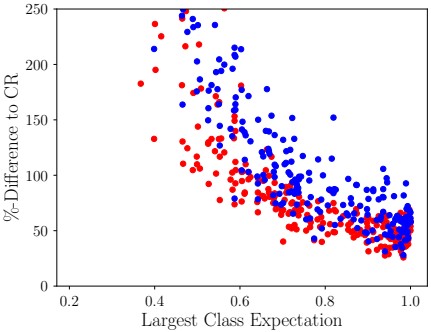

Figure 1: Percentage distance between constructed adversarial perturbations and the certified radii given by Equation 3, with Our technique in Red and PGD in Blue. Results for CIFAR-10 with $\sigma = 1.0$. Additional experimental results are visible in Figure 3.

large adversarial perturbations for MNIST may be driven by the polarised nature of these samples—containing regions of black and white with limited blending—and the constructed gradients leading to adversarial perturbations outside the numerical domain, which are then clipped to $[0, 1]^d$.

**Performance relative to certified guarantees**   An important feature of these results is the difference between the attack proportion of our technique, and the bound provided by Cohen et al. (Cohen et al., 2019). Intuitively it would be expected that the linear multiplicative proportionality of Equation (3) would increase the magnitude of the guarantee in $L_2$ norm space, and thus in turn introduce a greater difference between our attack and the certified guarantee. However, in practice these increases are offset by both a decrease in both the class expectations—due to the smoothing influence of additive noise—and the difficulty of constructing an attack.

In the context of the Cohen et al. bound of Equation 3, by Table 1 our technique clearly produces smaller adversarial perturbations for the majority of samples, while consuming approximately the same amount of computational time as both PGD and Carlini-Wagner. When compared against PGD, Figure 1 underscores the difference in the magnitude of the adversarial perturbation when considered against Cohen et al., with a clear self-similar trend in which the percentage difference to Equation (3) increases as the largest class expectation decreases in Figure 3. This suggests that the delta is likely not the attacks failing to identify global optimal adversarial perturbations in this region, but rather that this region would instead potentially admit larger certifications by taking a revised approach. There also appears to be a correlation between the outperformance of our approach and the semantic complexity of the prediction task, which suggests that tightening these guarantees could be increasingly relevant for complex datasets of academic and industrial interest.

**Limitations**   It is important to note that this current work has deliberately focused upon $L_2$-norm attacks as our technique is built upon certified guarantees of robustness, which are primarily built upon the potential for adversarial examples bounded in $L_2$-norm space and do not presently extend to rotational or translational modifications (Tian et al., 2018), nor functional attacks (Laidlaw & Feizi, 2019) attacks. While this inherently biases our approach towards datasets with image-structured data, the core concepts of attacking randomised smoothing, and of augmenting attack methodologies with the knowledge of regions of class invariance should be readily extensible to a broad array of data types and structures.

We also acknowledge that the chosen attacks exist at a single point of parameter space, and there are many other attacks that remain untested, which may provide different results. We emphasise here that the chosen set of attacks have distinct conceptual mechanisms, and were chosen to explore the relative performance of other gradient-based mechanisms relative to our new approach. While in some cases extended versions of these attacks exist, there is no guarantee that this increased complexity yields better results in the context of randomised smoothing.

Finally, this attack requires access to significant amounts of GPU memory. Attacking a ResNet18 model trained for CIFAR-10 required approximately 10 GB of GPU memory when smoothing was

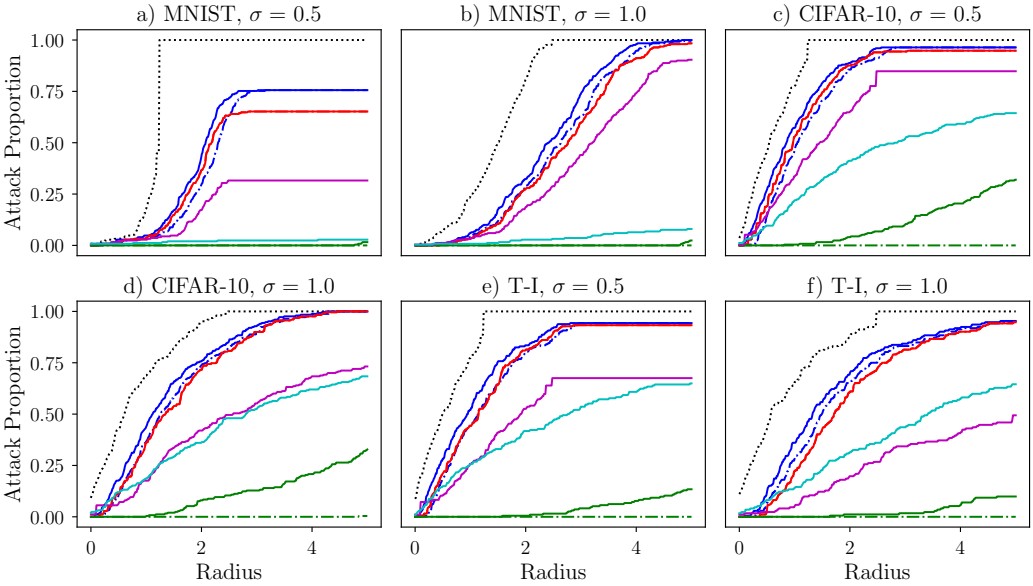

Figure 2: Best achieved Attack Proportion for our new Certification Aware Attack (Blue), PGD (Red), DeepFool (Cyan), Carlini-Wagner (Green), and AutoAttack (Magenta), with the black dotted line demonstrating the theoretical best case performance, as calculated by Cohen et al.. Dashed lines represent the first identified attack, while solid lines denote the best identified attack.

performed over 1500 samples, whereas close to 37 GB was required for Tiny-Imagenet with 1000 samples. This memory consumption is driven by our current implementation requiring all samples to be loaded into memory at once, prior to performing the gradient-based iterative step. While this process can be improved through batching, we chose not to follow this to ensure that our results weren't influenced by the batching implementation. Such a change would be necessary though to attack datasets containing larger images, or more memory-intensive model architectures.

## 5 CONCLUSION

While it is well known that adding calibrated noise to models via randomised smoothing can improve adversarial robustness, this work demonstrates that this process and the resulting certifications introduce a heretofore undiscovered attack surface, yielding our Certification Aware Attacks. By leveraging the guarantees of class invariance provided by randomised smoothing for both the correct and malicious class prediction, these attacks have the potential to significantly decrease the size of identified adversarial perturbations, relative to other techniques. Relative to tested baseline attacks, our new attack finds smaller adversarial perturbations for almost 90% of samples, resulting in a 13% reduction in the median attack perturbation. Decreasing the magnitude of these adversarial perturbation makes the attacks more difficult to detect, and thus, more likely to be successfully passed through a model.

## 6 ETHICS STATEMENT

While constructing such attacks has malicious benefits, there is also inherent value in understanding the potential of models to be compromised. As we discover within this work, certification mechanisms introduce a heretofore undiscovered attack surface that can be exploited to further decrease attack radii. As such, exploring such attacks is important to defray potential overconfidence regarding the robustness of certified models, and to provide contextual information about the values (or lack thereof) of deploying autonomous machine learning models in high stakes environments.

## 7 REPRODUCABILITY STATEMENT

This work describes both a general framework for attacking models defended by randomised smoothing, and a specific attack (as described in Algorithm 1) that can be used to attack such models. To enhance reproducability, prototype code has been attached to the OpenReview submission, and will be released publicly upon publication.

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

---

**Algorithm 2** Class prediction and certification for the Certification Aware Attack Algorithm of Algorithm 1.

---

1: **Input:** Perturbed data $\mathbf{x}'$, samples $N$, level of added noise $\sigma$
2: $\mathbf{y} = \mathbf{0}$
3: **for** i = 1:N **do**
4: $\quad y_j = y_j + 1$ if $GS\left(f_\theta\left(\mathbf{x}' + \mathcal{N}(0, \boldsymbol{\sigma}^2)\right)\right) = j$ {Here $GS$ is the Gumbel-Softmax}
5: **end for**
6: $\mathbf{y} = \frac{1}{N}\mathbf{y}$
7: $z_0, z_1 = \text{topk}(\mathbf{y}, k = 2)$ {topk is used as it is differentiable, $z_0 > z_1$}
8: $\breve{E}_0, \widehat{E}_1 = \text{lowerbound}(\mathbf{y}, z_0), \text{upperbound}(\mathbf{y}, z_)1$ {Calculated by way of Goodman et al. (Goodman, 1965)}
9: $R = \frac{\sigma}{2}\left(\Phi^{-1}(\breve{E}_0) - \Phi^{-1}(\widehat{E}_1)\right)$
10: **return** $\breve{E}_0, \widehat{E}_1, R$

---

# A    APPENDIX

# B    ALGORITHMIC DETAILS

Algorithm 2 outlines the steps sampling process required to generate both the class prediction and the certification radius, which is expressed in terms of the lower and upper bounds of the largest and second largest class, respectively labelled as $\breve{E}_0$ and $\widehat{E}_1$. These expectations are calculated by way of a Monte-Carlo approximation of $\mathbf{y} = E[f_\theta(\mathbf{x}'; \sigma, N)]$, subject to the application of a concentration inequality to quantify the underlying uncertainties.

# C    ATTACK CONFIGURATION

Following previous experimental works, we employed the following hyperparameters for each attack framework. For **Carlini-Wagner**, we set that $\kappa$ of Equation 2 to 0, and weighted the loss from the one-hot encoding by $10^{-4}$. The Carlini-Wagner training process was conducted using a learning rate of 0.01 over 100 iterations. Similarly **DeepFool** also employed 100 iterations, and employed an overshoot factor of 0.02. When considering **PGD** of the form outlined in Equation 1 the $\epsilon$ was set at $\frac{20}{255}$, with iterations again occurring 100 times. Further details relating to the choice of $\epsilon$ for PGD can be found within Appendix E.4.

**AutoAttack** was performed using the randomised model variant, with the attack radii set at $\max(2 \times R, 0.1)$, where $R$ was calculated by Equation 3. This choice of attack radii was a deliberate attempt to match the average identified attack radii (relative to Equation 3), as is further discussed in Appendix E.5.

# D    PERFORMANCE OF INDIVIDUAL SAMPLES

When comparing the individual performance of samples, as is seen in Figure 3 , it is clear that there's a consistent and marked improvement across the suite of experiments. The one exception to this is when the largest class expectation is 1 and $\sigma = 0.5$, which leads to an interesting and marked increase in the radii produced by our technique, relative to that of PGD. This appears to be a product of how we estimate the bounds on the two highest class expectations, prior to calculating the iterative step. By underestimating the uncertainties in this narrow region, we introduce significant growth in the predicted certified radii, which in turn induces a significant overestimate in the radius of certification. This in turn induces oversized iterative steps, which prevent converging upon smaller potential adversarial perturbations.

That these graphs show no examples with percentage differences less than 25% also reinforces our earlier point about there still being potential for improving certified guarantees. This is especially true as the largest class expectation decreases, as the minimum observed percentage difference begins to increase significantly.

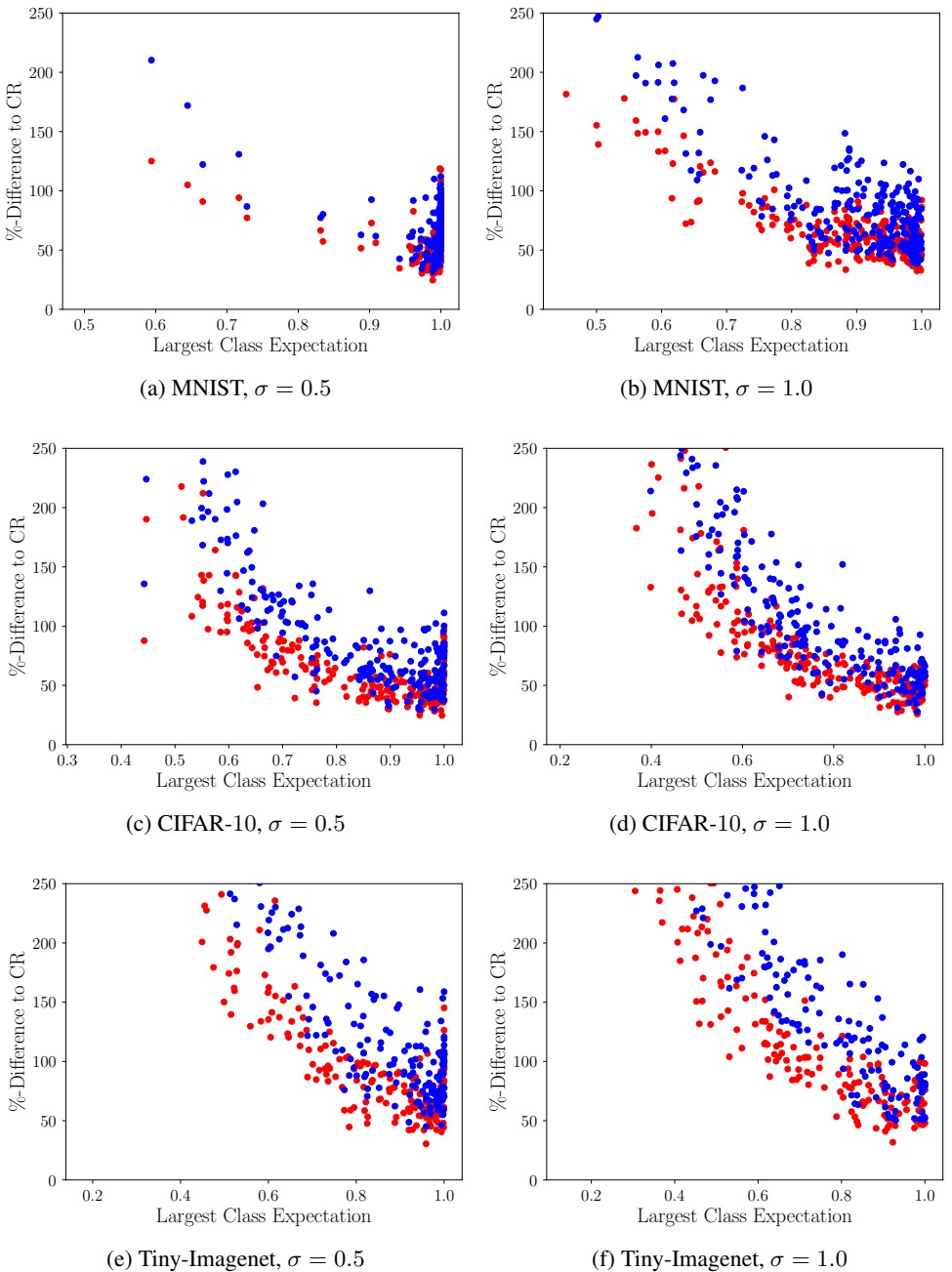

Figure 3: Per-sample performance of our Certification Aware Attack technique (Red) and PGD (Blue), relative to the magnitude of the $L_2$ norm guarantees provided by Certified Robustness.

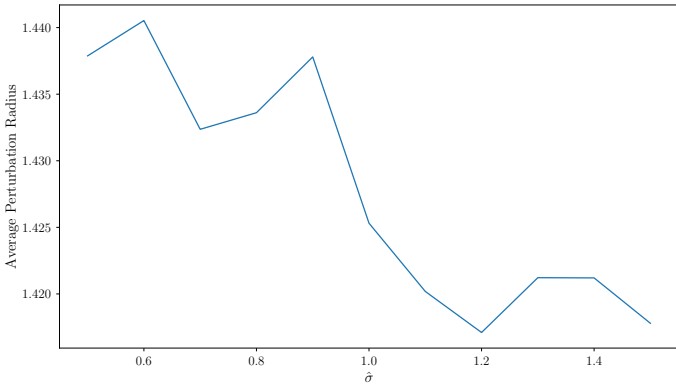

Figure 4: Average perturbation radius when $\sigma$ is estimated by $\hat{\sigma}$. Data collected for CIFAR-10 at $\sigma = 1.0$.

Table 2: Evolution of metrics as a function of the number of samples used to estimate the expectations, as calculated for CIFAR-10 with $\sigma = 0.5$. All other experiments in this paper were calculated using 1500 samples for MNIST and CIFAR-10, and 1000 samples for Tiny Imagenet. AutoAttack experiments were excised from these experiments due to the underlying computational cost.

| Method | Metric | 250 | 500 | 750 | 1000 | 2000 |
|---|---|---|---|---|---|---|
| Ours | $r_{50}$ | 0.94 | 0.84 | 0.85 | 0.82 | 0.79 |
| | Attacked | 96% | 97% | 97% | 99% | 98% |
| | Time (s) | 1.2 | 1.6 | 2.6 | 4.5 | 7.1 |
| PGD | $r_{50}$ | 1.04 | 0.94 | 0.89 | 0.9 | 0.87 |
| | Attacked | 95% | 96% | 96% | 96% | 96% |
| | Time (s) | 0.6 | 0.8 | 1.1 | 1.4 | 1.9 |
| C-W | $r_{50}$ | 6.41 | 6.45 | 6.46 | 6.41 | 6.34 |
| | Attacked | 93% | 92% | 93% | 94% | 94% |
| | Time (s) | 1.9 | 2.8 | 4.0 | 5.2 | 7.4 |
| DeepFool | $r_{50}$ | 2.18 | 1.91 | 1.83 | 1.84 | 1.81 |
| | Attacked | 76% | 82% | 81% | 81% | 85% |
| | Time (s) | 0.3 | 0.4 | 0.7 | 0.9 | 1.1 |

# E    SENSITIVITY ANALYSIS

## E.1    ACCURACY OF $\sigma$

The white-box threat model assumes that the attacker has access to the full model and its parameters, including the level of additive noise $\sigma$. However, for the purposes of constructing the attack, $\sigma$ is only required for constructing the maximum step-size through Equation 11, which is employed alongside the cutoff factor $c$ and scaling factor $m$. That $m$ and $c$ are scaling $\sigma$ implies that accurate knowledge of $\sigma$ is in fact not strictly necessary. In fact, as shown by Figure 4, even over-estimating $\sigma$ by 50% decreases the radius of the identified adversarial perturbation. While this strongly suggests that there is further scope for optimising $m$ and $c$, it also makes it clear that even estimating $\sigma$ as part of a surrogate model, in order to attack under a black-box threat mode.

## E.2    SAMPLE SIZE

In order to assess the influence of the sample size on the relative performance of the techniques, Table 2 considers the performance of the best identified adversarial example. Due to the influence of the sample size on the measured uncertainties, there is a small but steady decline in the radii of observed adversarial examples with the sample size. Surprisingly, increasing the samples did induce a slight increase in the proportion of adversarial perturbations identified, suggesting that the

Table 3: Comparison of the influence of the cutoff $c$ in Our attack process for $\sigma = 0.5$, in terms of the median first and best adversarial example, and the median time for identifying the first adversarial example.

| Dataset | Cutoff $c$ | Successful | First $r_{50}$ | Best $r_{50}$ | First $t_{50}$ (s) |
|---------|-----------|-----------|----------------|---------------|---------------------|
| MNIST | 0.05 | 45.0% | 1.646 | 1.617 | 4.44 |
| | 0.25 | 65.0% | 1.988 | 1.874 | 1.51 |
| | 0.5 | 71.5% | 2.149 | 1.942 | 1.16 |
| | 0.75 | 75.0% | 2.345 | 1.962 | 1.07 |
| | 1.0 | 75.5% | 2.426 | 1.972 | 1.07 |
| | 1.5 | 74.5% | 2.687 | 1.993 | 0.97 |
| | 3.0 | 74.0% | 3.257 | 2.020 | 0.97 |
| CIFAR-10 | 0.05 | 93.5% | 0.806 | 0.756 | 1.76 |
| | 0.25 | 97.5% | 0.904 | 0.801 | 0.74 |
| | 0.5 | 97.0% | 1.030 | 0.807 | 0.65 |
| | 0.75 | 97.0% | 1.135 | 0.814 | 0.65 |
| | 1.0 | 99.0% | 1.220 | 0.838 | 0.65 |
| | 1.5 | 98.5% | 1.486 | 0.829 | 0.65 |
| | 3.0 | 99.0% | 2.194 | 0.845 | 0.65 |

smoothing influence of increasing the sample count may be making the gradient trajectories slightly more amenable to identifying adversarial perturbations. A curious feature of these results is that while the increases in computational time are sub-linear with respect to the number of samples for all techniques except our Certification Aware Attacks. This suggests that additional improvements in our technique's performance could likely be achieved with additional code optimisation.

### E.3 CUTOFF FACTOR $c$

It is crucial to understand the influence of the cutoff factor $c$ from Equation (11), due to its centrality within Algorithm 1. In order to do so Table 3 considered the sensitivity of the MNIST and CIFAR-10 experiments to changes in $c$. Specific focus was placed upon the case where $\sigma = 0.5$, as the label space for the lower level of noise is less smoothed, and thus should be more sensitive to the maximum step-size. While we have excised Tiny-Imagenet due to computational cost concerns, for MNIST and CIFAR-10 it is clear that beyond $c = 0.5$ there is effectively no advantage in increasing the cutoff factor further, when both the converged adversarial example, and the rate of convergence. The decreased sensitivity to $c$ beyond this point is a consequence of the distribution of likely certified radii—specifically that as $c$ increases it becomes increasingly less likely that a sample point would lead to a certification of a radii larger than this value.

### E.4 PGD STEP-SIZE $\epsilon$

Across all our experiments, the step-size for PGD was uniformly set to $\epsilon = 20/255$, primarily as this parameter matched the results of prior works attacking these datasets in the absence of randomised smoothing, and such attacks are likely to be the primary source of information for an attacker attempting to modify their approach to consider defended models. However, Figure 5 does demonstrate that decreasing $\epsilon$ below this level will yield decreased perturbation radii, doing so would further increase the computational cost of PGD relative to our new technique, while still yielding an insignificant change in the number of samples for which PGD yields the smallest certification.

### E.5 AUTOATTACK RADIUS $\epsilon$

As was discussed in Section 2, AutoAttack's $\epsilon$ is not the step-size of the iterative process, but rather the maximum *bound on the norm of adversarial perturbations*. While theoretically this $\epsilon$ should serve as an upper bound, in practice our experiments showed that the significant majority of identified adversarial examples were produced with a radii of $\epsilon$. To explore the performance of changes in how $\epsilon$ is specified—either directly or as a multiple of the Cohen certified radius of the original sample

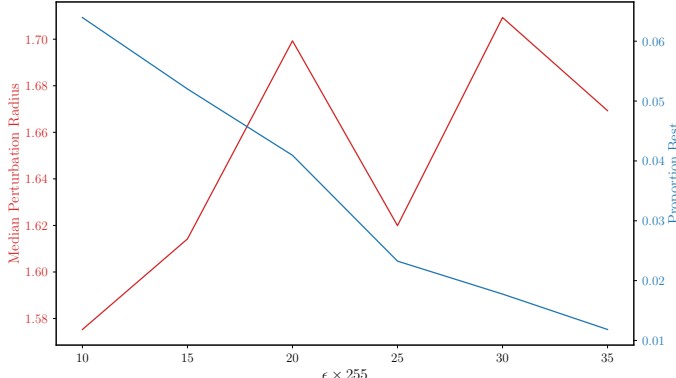

Figure 5: Performance of PGD for varying the step-size $\epsilon$. Results for CIFAR-10 at $\sigma = 1.0$.

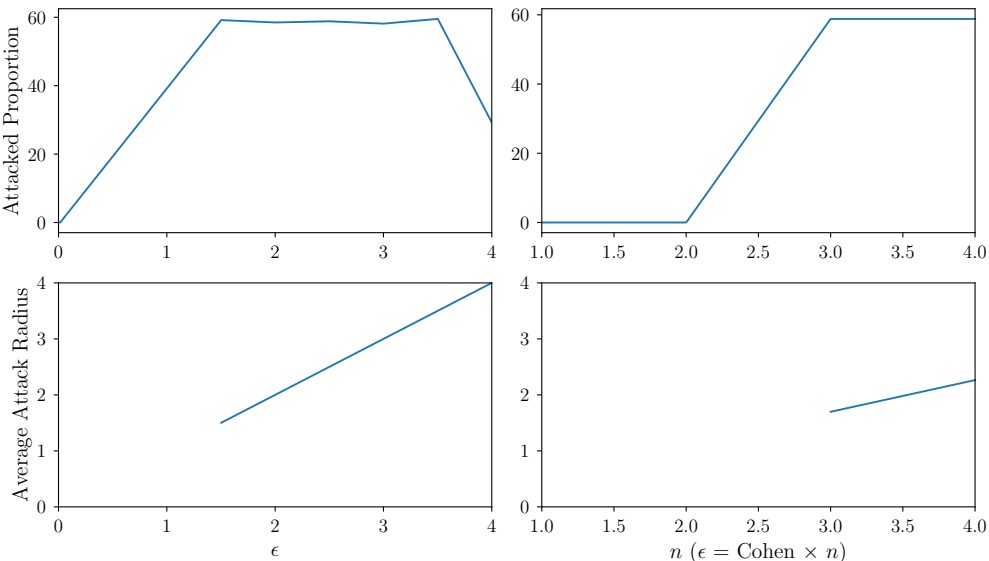

Figure 6: Performance of AutoAttack for varying $\epsilon$, where $\epsilon$ is either set directly or as a multiple of the Cohen certified radius. Results for CIFAR-10 at $\sigma = 1.0$.

point—Figure 6 demonstrates the relative performance of AutoAttack. While increasing the radius of certification does increase the proportion of samples that are able to be viably attacked, we uniformly set $n = 2$ for all test sets to match the average size of the adversarial perturbation identified by our technique (relative to the certification provided by Cohen).

## F  TRAINING WITH MACER

Recent work has considered how the certified proportion of models can be improved by augmenting the training reward to maximising the expectation gap between classes Salman et al. (2019). A popular approach for this is MACER Zhai et al. (2020), in which the training loss is augmented to incorporate what the authors dub as the $\epsilon$-robustness loss, which reflects proportion of training samples with robustness above a threshold level. Such a training time modification can increase the average certified radius by $10 - 20\%$, however doing so does increase the overall training cost by more than an order of magnitude.

To test the performance of our new attack framework against models trained with MACER, Table 4 and Figure 7 recreate earlier results from within this work. Under such training time modifications our approach produces both consistent attack radii—relative to Table 1—and the same levels of

Table 4: Performance metrics for MNIST (M), Cifar-10 (C), and Tiny-Imagenet (TI) for varying $\sigma$. 'Success' and 'Best' are the proportion samples for which each attack was success, and outperformed all others. $r_{50}$ and %-Cohen are the median attack and the size relative to the guarantee of Cohen.

| Categorisation | | Smallest Attack | | | | | First Attack | |
| Data | Attack | Success | Best | $r_{50}$ | %-Cohen | Time (s) | Ratio($r_{50}$) | Ratio(Time) |
|---|---|---|---|---|---|---|---|---|
| C-0.25 | Ours | 100% | 100% | 0.91 | 1188% | 17.54 | 1.20 | 0.35 |
| | PGD | 100% | 0% | 15.17 | 20977% | 26.25 | 1.00 | 0.02 |
| | C-W | 96% | 0% | 10.61 | 13958% | 4.04 | 1.30 | 0.08 |
| | D.Fool | 100% | 0% | 2.10 | 3105% | 9.59 | 1.00 | 1.00 |
| C-0.5 | Ours | 92% | 92% | 1.56 | 1148% | 19.33 | 1.08 | 0.55 |
| | PGD | 100% | 0% | 15.58 | 11912% | 26.32 | 1.00 | 0.02 |
| | C-W | 87% | 2% | 11.58 | 7242% | 3.37 | 1.23 | 0.10 |
| | D.Fool | 100% | 5% | 4.28 | 3353% | 11.51 | 1.00 | 1.00 |
| C-1.0 | Ours | 75% | 75% | 2.03 | 1209% | 19.57 | 1.05 | 0.66 |
| | PGD | 100% | 1% | 16.14 | 8101% | 26.24 | 1.00 | 0.02 |
| | C-W | 99% | 11% | 11.29 | 5932% | 3.36 | 1.24 | 0.10 |
| | D.Fool | 100% | 13% | 7.49 | 3984% | 9.61 | 1.00 | 1.00 |

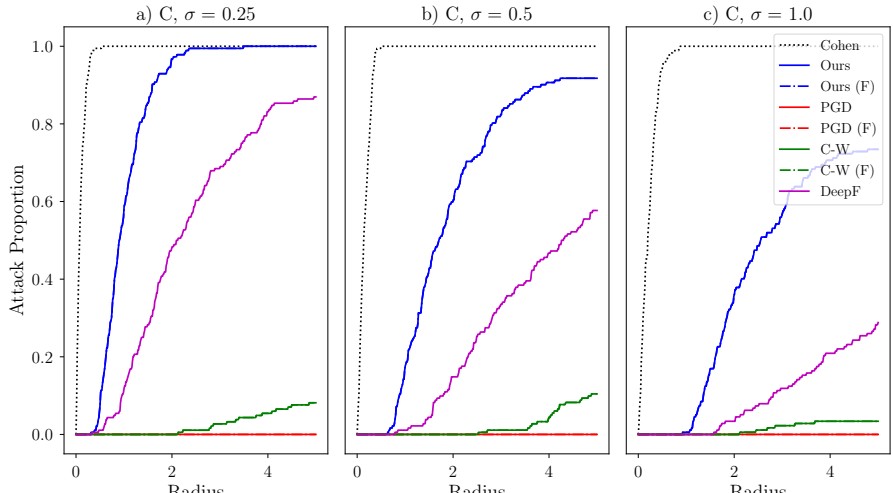

Figure 7: Attack and Certification Performance for a Resnet-110 model for Cifar-10, when trained with MACER. Similar to Figure 2, an ideal attack will approach the Cohen radii suggested by the black dotted lines.

out-performance that were seen in the earlier results. However, we must emphasise that the percentage difference to the certified Cohen radius for all techniques has significantly increased, which is a consequence of the average certified radius *decreasing four-fold* under the MACER training routine. This appears to be a consequence of the larger model architecture in these tests, with these latter tests employing a Resnet-110 architecture, as compared to Resnet-20 in the main body of the paper. This decrease in the average certification increases the average computational time for our approach, as a greater number of iterative steps are required to converge upon a solution, as the step size is proportional to the calculated certified radius.

