# OpenReview forum: "Exploiting Certified Defences to Attack Randomised Smoothing"
_ICLR.cc/2023/Conference — Submitted to ICLR 2023_

### Official Review · Reviewer_Ms23 · 2022-10-23

**Confidence:** 4
**Correctness:** 4
**Technical Novelty And Significance:** 2
**Empirical Novelty And Significance:** 2
**Recommendation:** 5

**Clarity, Quality, Novelty And Reproducibility:**

The paper is reasonably well written. At times, the paper's choice of notation clashes with that commonly used in the literature which is a bit annoying (e.g., in the existing literature, \epsilon *always* refers to the perturbation size, and not to the step-size)
The paper's novelty is the main limiting factor, and the scope is quite limited: the paper focuses on minimal-norm adversarial examples against randomized smoothing. There are many works on minimal-norm attacks. There are also many works on attacking randomized defenses. It isn't quite clear what new things this paper brings to the topic.

**Strength And Weaknesses:**

- The paper is not particularly well motivated: the whole point of certified defenses is that we get a proof of robustness so optimizing attacks against these models is not necessarily super interesting (unless maybe the point is to check how tight the certification guarantees are).
- The empirical comparison mainly considers prior attacks that were developed for deterministic defenses (with the exception of autoattack). So naturally such attacks will not perform well against randomized smoothing.


**Summary Of The Paper:**

The paper considers empirical attacks against models protected with randomized smoothing.
The paper proposes some simple optimizations and shows to find smaller adversarial examples than other attacks.

**Summary Of The Review:**

My recommendation is mainly due to a perceived lack of novelty and motivation for the current paper.
The scope is very limited, and within this limited scope there are many alternative baselines that could be considered.

For example, all the considered attacks should be combined with some form of expectation-over-transformation (EOT) since the defense is randomized. Moreover, instead of using a gumble-softmax trick, it seems a lot simpler to just run a "multi-targeted" attack (https://arxiv.org/abs/1910.09338) that targets each possible class in term.

Ultimately, even if the evaluation were extended to better compare against these attacks, it remains unclear what to make of these results. Certified defenses are certified to be robust! And many existing works already show that these robustness guarantees are close to tight (i.e., if a randomized smoothing defense has a certified accuracy of say 50% for perturbations of size 1, then we already know empirically that the best attack with that perturbation budget will bring the defense's accuracy down to roughly 50%).
This paper instead considers minimal-norm attacks, but it isn't clear why this distinction matters. Certified defenses aim at maximizing robust accuracy at a fixed perturbation level \epsilon. They don't aim to make any guarantees that successful perturbations will be any larger than \epsilon.

---

> ### Author Response · Authors · 2022-11-10
> **Response to reviewer Ms23**
>
> To begin with, we thank the reviewer for their time and expertise.
>
> We do agree that there are many papers on attacking randomised-defences, it is the very existence of such attacks that have, in part, motivated the development of certified defences. Within this work we produce the first minimal-norm attack against such certified models, demonstrating both that such attacks are possible, and more importantly, that the very nature of the certification can be exploited to construct adversarial examples with smaller perturbation sizes. The existence and construction of such minimal-norm attacks is important, as it has been noted (by Gilmer et al (2018), as discussed in Section 3.3) that decreasing the norm of attacks can lead to decreased detectability of attacks.
>
> When it comes to certification, we must emphasise that while this reviewer is correct that a randomised smoothing model with a given certified accuracy can be used as a basis for understanding **aggregate** features of a models behaviour, it says nothing about the behaviour of **individual samples** under attack. We would strongly argue the idea that "certified defences are certified to be robust" and that  "the whole point of certified defenses is that we get a proof of robustness" to be a misleading view of the certification space, as the level of provable robustness for an individual sample is often so small that there is still significant scope for small adversarial perturbations to influence the behaviour of individual samples. The level of robustness provided is not enough to guarantee that no adversarial examples exist, nor that any such adversarial perturbation would be large enough to lead to it being trivially detected. Within this very work we show - through the median achieved attack radius of Table 1 - that such adversarial examples exist, and they are small enough that their existence should still be considered as a threat to deployed models, even if these models have an associated certification. This is especially true in light of our core observation: that the very nature and existence of certifications for an individual sample makes it easier to construct attacks against said sample.
>
> Regarding expectation over transformation and the comment that our work only "considers prior attacks that were developed for deterministic defenses...so naturally such attacks will not perform well against randomized smoothing" - we would emphasise that there is no need to deploy EOT to make the models outputs deterministic (in a manner aligned with the typical use case of the comparison attacks), as our attacks are being deployed not against individual draws under noise, but rather against the expected output of the model, which is approximated by a very highly concentrated quantity that is essentially deterministic.
>
> With regards to the suggestion of deploying multi-targeted attacks, rather than the gumbel-softmax trick, while we acknowledge we may not be fully following the reviewers point, but we would like to emphasise that
> 1. The gumbel-softmax is applied because we are attacking E[F(x + N) = i] - the expected output of the model F to input x perturbed by noise N, where the final layer of F outputs an argmax. As derivatives of the argmax cannot be calculated, we employ the gumbel-softmax to make differentiating this process possible.
> 2. It appears that the multi-targeted attacks approach has been designed around L_inf attacks, whereas our approach considers L_2 norm perturbations, as the latter space is the most commonly considered by mechanisms of certified robustness.
>
> We hope that this has helped clarify the points raised within this review, and emphasise our willingness to engage in future dialogue.

---

> ### Author Response · Authors · 2022-11-18
> **Reminder of response**
>
> With ~12 hours to go in the discussion period, we would just like to reiterate that we believe the reviewers statements regarding the nature of certification have only considered the aggregate performance of such models, rather than the potential to attack individual samples. A model producing a robustness certificate does not make it robust to all attacks. Rather each sample can be certified up to a radii (with a different radii for each sample), and these radii are typically small enough to admit human imperceptible adversarial attacks, or attacks that are potentially small enough to evade detection methodologies. In this work we showed that the size of the identified adversarial perturbations could be decreased when exploiting the certified radii of points to guide the attack process, which is a heretofore undiscovered attack surface.
>
> We also again emphasise that these attacks were performed not against individual draws performed by randomised smoothing, but the expected output of the model, thus obviating the need for employing techniques like EOT.

---

### Official Review · Reviewer_YX18 · 2022-10-25

**Confidence:** 3
**Correctness:** 3
**Technical Novelty And Significance:** 2
**Empirical Novelty And Significance:** 2
**Recommendation:** 3

**Clarity, Quality, Novelty And Reproducibility:**

The methodology is generally clearly presented. The technical novelty is questionable as it is in some sense a simple extension of SmoothAdv [Salman et al., 2019] in flavor of CW-attack.

**Strength And Weaknesses:**

**Strength**

* The paper is easy-to-follow
* An extensive experiment is performed to confirm the effectiveness


**Weakness**

* Overall, I feel I do not understand the motivation of the paper - In what practical scenarios that one can be interested in finding smallest adversarial examples for certifiably robust models, i.e., smoothed classifiers? Providing more examples on concrete applications of this technique would be really helpful to get readers more motivated. For example, could the adversarial examples found from this method transfer to other (possibly non-smoothed) models? Also, as another example, the paper could perform an in-depth analysis from the found adversarial examples to broaden understanding of smoothed classifiers.
* The paper should discuss and compare the proposed method with SmoothAdv [Salman et al., 2019] which also propose an adversarial attack specialized for randomized smoothing. Also, could this method be used to improve SmoothAdv, i.e., to improve adversarial training of smoothed classifiers?
* The paper lacks on ablation study on the proposed components.



**Summary Of The Paper:**

The paper designs an adversarial attack method that acts on randomized smoothing based models, which in fact provide provable guarantee on adversarial robustness. Specifically, the paper approximates the non-differentiable operation of randomized smoothing with Gumbel-softmax, and a CW-like attack objective, and a heuristic on the attack step-size based on the sample-wise certified radius given by Cohen et al. (2019). Experimental results on MNIST, CIFAR-10 and Tiny-ImageNet show that the proposed method can find a smaller adversarial examples from a smoothed classifier compared to other empirical attack methods.

**Summary Of The Review:**

I generally feel from the paper a lack of motivation for the proposed method. Specifically, for now I am not quite convinced on a demand for a stronger attack method against certifiably robust classifiers. In terms of evaluation, I think the paper should compare with SmoothAdv [Salman et al., 2019] to claim its technical novelty.

---

> ### Author Response · Authors · 2022-11-09
> **Response to reviewer YX18**
>
> We thank reviewer YX18 for the time they placed into this review.
>
> To begin by addressing your first weakness regarding the overall motivation: the need for minimal-norm adversarial examples has been well documented in the literature. As was noted in the first paragraph of Section 3.3, finding smaller adversarial examples has been previously established (see Gilmer et al, 2018) to be correlated with a decreased likelihood of detection. Indeed the entire motivation for certification is to qualitatively demonstrate the lack of existence of **small** adversarial examples. This work demonstrates that certified models admit an additional attack surface, which allows the size of constructed attacks to be reduced, producing attack sizes that are significantly smaller than SOTA. We also emphasise that a certifiably robust model does not guarantee that no adversarial examples exist, but rather that on a sample-by-sample basis certifications provide guarantees that no adversarial examples exist within a fixed radius. As is shown in Figure 2, our new attack identifies attacks that are closer to that radius than would be achieved by other works.
>
> Relating to your question about transferability - it is highly likely that any adversarial example against a model with a certified defence would also serve as an adversarial example against the un-smoothed model, due to the nature of the smoothing process. However, we emphasise that our specific concern was models that present as providing certified robustness, and constructing attacks against such models. We see no circumstances under which such a model would be attacked for any reason other than to attack the model itself, rather than as the basis for any transferability.
>
> Regarding SmoothAdv of Salman et. al - SmoothAdv certifies using the same Cohen equations that we deployed within our work. SmoothADV is a training time modifications to the certified radius, and as our approach is independent of the training time approach taken. However in response to both yourself and another reviewer, we have uploaded a revised version of the paper which includes the MACER training time modifications (of Zhai et. al, 2020), which have now been included as Appendix F. These results demonstrate that our technique still **significantly outperforms the other attack frameworks** when deployed against a certified model trained with MACER for multiple levels of added noise for a Cifar-10 model trained with a Resnet-110 architecture.
>
> Finally, relating to your comment that "the technical novelty is questionable as it is in some sense a simple extension of SmoothAdv [Salman et al., 2019] in flavor of CW-attack" - while it is true that Salman exploits the existence of adversarial attacks for models defended by randomised smoothing, their approach involves finding the existence of an adversarial example to guide a **training time** process; whereas our approach attacks **test time** samples not by a CW-attack, but by exploiting the nature of certifications to guide the step size control and objectives. If we were simply performing a CW-attack, then we would not observe the significant outperformance that our technique achieves relative to CW. The technical novelty is: the very nature of certified robustness admits a new threat surface, which makes it easier to attack certified models than had previously been thought. We believe that this discovery is at least as important as any technical contribution as it improves SOTA significantly with a simple attack and opens up a new research direction to further exploit a new attack surface introduced by certified robustness itself.
>
> We hope that this helps clarify the motivation behind this work, and would welcome any additional questions.

---

> ### Author Response · Authors · 2022-11-18
> **Reminder of response**
>
> We would just like to take this time to reiterate that our rebuttal revision has incorporated experiments against the MACER training time modification (which is similar to SmoothAdv), which has been placed in Appendix F. These results show that changing the training mechanism of certified models does not influence the performance of our technique for attacking individual samples in deployed models, by exploiting the additional attack surface that certified robustness introduces.
>
> We would also emphasise that the same motivation for finding minimal norm attacks against general models applies to certifiably robust models - in that minimal norm attacks are harder to detect, and have the potential to attack more samples. The fact that these models are certifiably robust does not guarantee a resistance to adversarial attacks, especially when the size of the guarantees provided are well below the level of human perceptability for most samples.

---

### Official Review · Reviewer_rMga · 2022-10-25

**Confidence:** 4
**Clarity, Quality, Novelty And Reproducibility:** See above.
**Correctness:** 3
**Technical Novelty And Significance:** 3
**Empirical Novelty And Significance:** 2
**Recommendation:** 5

**Strength And Weaknesses:**


Strength:
1) Propose a novel method to attack randomized smoothing/
2) Achieve  state-of-the-art performance compared with previous attack methods.
Weakness:
1) Not enough experiments to support the statements:
a. Only evaluate on randomized smoothing model of Cohen19. The performance of the attack on other randomized smoothing methods (such as SmoothAdv, MACER, Carlini22:https://arxiv.org/abs/2206.10550) is unclear.
b. The setting of the /sigma in Table 1 is not aligned with Cohen19. Experiments under smaller /sigma like 0.12 or 0.25 are not shown.
c. Experiments are limited to specific model architecture and datasets. Model architecture ResNet18 and Dataset Tiny-ImageNet are simple compared with Cohen19’s.

2) Ambiguous explanation of evaluating metrics of the experiments:
Do not have an explicit explanation of the evaluation metrics in Table 1 like Success, Best, r_50, %-Cohen, which makes me confusing.

3) Questions about AutoAttack
As I know, AutoAttack ensembles 4 attack methods which include APGD-CE (PGD with adaptive step size) and APGD-DLR (similar to target CW attack). Why does AutoAttack have a worse performance than PGD and CW?


**Summary Of The Paper:**

This paper proposes a method to attack against models defended by randomized smoothing, and identify smaller adversarial perturbations for smoothed classifier than previous methods.


**Summary Of The Review:**

Novel task and attack method under certified robustness via randomized smoothing, but considering lack of experiments and detailed explanation, I could not recommend this paper for acceptance at this moment.

---

> ### Author Response · Authors · 2022-11-09
> **Response to reviewer rMga**
>
> Thank you to reviewer rMga for the time and knowledge they placed into this review. To address your listed weaknesses in the order in which they were presented:
>
> 3) a) We emphasise that techniques like SmoothAdv and MACER both use Cohen style certification under the hood (they are simply training time modifications that seek to enhance the achieved certified radii), and as such any attack vector that Cohen introduces can similarly be exploited against such models. In response to your comment we have uploaded a revised version of the paper which includes the MACER training time modifications (of Zhai et. al, 2020), which have now been included as Appendix F. These results demonstrate that our technique still **significantly outperforms the other attack frameworks** when deployed against a certified model trained with MACER for multiple levels of added noise for a Cifar-10 model trained with a Resnet-110 architecture.
>
> b) The focus upon higher levels of sigma was motivated by the observation that smaller values of sigma yield such small certifications that they're unlikely to be used in any practical deployed system. However, as part of the additions within Appendix F, we have also tested against sigma = 0.25, which shows an even greater level of out performance by our technique, relative to the compared approaches. We also emphasise here that we are not attempting to match or compare to Cohen19, but rather to demonstrate that certifications---like those seen in Cohen19, SmoothADV, and MACER---all admit an additional, heretofore undiscovered attack surface, which allows for the construction of minimal-norm attacks against such models.
>
> c) The choice of ResNet18 and Tiny-ImageNet are products of limitations in access to computational resources, however we emphasise that Tiny-Imagement is still orders of magnitude more complex than other reference datasets, and that the existence of the attack surface---our core observation---is independent of the model or dataset. However to help emphasise the model independent nature of this attack surface, Appendix F was performed against a Resnet110 architecture.
>
> 4) We emphasise that these metrics are all explained within the caption of Table 1.
>
> 5) While this was alluded to in paragraph 4 of Section 2 and E.5, to be more clear - while PGD and CW allow the step size to be manually set, within AutoAttack the maximum step size is set, with the algorithm handling adjustments to the step size across iterative steps. When deployed against models defended by randomised smoothing the adaptive step sizing process that is at the core of AutoAttack appears to fail, leading to large computational times and a broad failure to converge.
>
> We once again thank the reviewer for their comments, and would welcome any additional dialogue.

---

> ### Author Response · Authors · 2022-11-18
> **Reminder of response**
>
> We'd just like to remind the reviewer that the rebuttal revision has been updated to include Appendix F, which covers experiments against MACER, which shows that the technique performs equally well in such a context. This is unsurprising as MACER and SmoothAdv are both training time modifications to the process of generating certifiably robust models, and these modifications do not influence the fundamental observation of our work: that the certified robustness of individual samples can be exploited to guide the construction of minimal norm adversarial examples, creating a new threat surface.

---

### Official Review · Reviewer_5baj · 2022-11-01

**Confidence:** 4
**Correctness:** 3
**Technical Novelty And Significance:** 2
**Empirical Novelty And Significance:** 1
**Recommendation:** 3

**Clarity, Quality, Novelty And Reproducibility:**

The paper uses very convoluted language and terminology at times, which makes it difficult to read and requires rereading some passages in the main paper. The authors could improve and simplify the presentation of the ideas. Overall the paper’s idea is novel but the technical novelty is limited. To improve the novelty, the authors should include additional experiments with other certified defenses to show that the proposed method can be useful and applied to other types of defenses.

**Strength And Weaknesses:**

### Strengths

- Proposed certification-aware adversarial attack, which improves the attack’s accuracy in comparison with existing attacks.
- A new way to adjust the perturbation step size based on the certification certificate.

### Weaknesses

- The paper considered only randomized smoothing and wasn’t applied to any other adversarial training or certified defenses.
- The comparison might be skewed in favor of the proposed method since the standard adversarial attacks are not aware of the randomized smoothing. Adding the expectation of transformation [1] and proper tuning of the attack’s parameters might change the results of the comparisons. Based on the provided experimental details, it is not clear if the authors conducted necessary hyperparameter tuning for other attacks.

[1] Athalye, A., Engstrom, L., Ilyas, A., & Kwok, K. (2018). Synthesizing Robust Adversarial Examples. In J. Dy, & A. Krause, Proceedings of the 35th International Conference on Machine Learning (pp. 284–293). Stockholmsm\"assan, Stockholm Sweden: PMLR.

**Summary Of The Paper:**

The authors proposed a certification-aware attack on randomized smoothing defense. To compute the gradient through non-differentiable randomized smoothing defense, they replaced arg max layers with the Gumbel-Softmax layer. Additionally, using the certification certificate, the perturbation step size can be automatically adjusted during the course of the attack, which is a novel idea for adversarial attacks. In the experiments, they compared their method with PGD, Carlini&Wagner, AutoAttack, and DeepFool attacks. Overall, the method improved the attack’s efficacy against randomized smoothing defense.

**Summary Of The Review:**

The authors proposed a Certification Aware adversarial attack, however, the proposed attack was only tested against randomized smoothing defense. The authors should have tested the attack against other certified defenses and empirical defenses. Due to the limited technical novelty and lack of detailed experimental comparison, my suggestion for this paper is to reject it. Nevertheless, the idea is quite interesting and promising.

---

> ### Author Response · Authors · 2022-11-09
> **Response to reviewer 5baj**
>
> We thank the reviewer for their comments, and would like to take the time to address the raised concerns.
>
> To begin with the weaknesses identified within the review:
> 1) The reviewer is correct in that we only considered randomised smoothing. This was due to its relative popularity within the certification literature, and the computational scaling issues that are present within other approaches (like those used within automatic bound propagation approaches like Auto-LiRPA and CROWN). Because of the popularity of randomised smoothing, we considered that demonstrating the presence of an additional attack surface---one that applies to all certified models---against randomised smoothing would be the most judicious use of the available space. However, we point the reviewer to the new Appendix F of the revised rebuttal submission, which contains experiments performed against MACER (of Zhai et. al, 2020), which is an alternate training and certification approach. Under such a framework, our model still significantly outperforms the compared attacks.
>
> 2) Regarding your concerns about the randomised nature of the models, and the potential value of incorporating expectations over transformation - we emphasise here that the attacks are not being deployed against an arbitrary randomised model (for which EOT may be a valid consideration), but rather **the attacks are being deployed against the expected output of the model**, which is approximated by a very highly concentrated quantity that is essentially deterministic. Our goal is to attack the most commonly used certification framework - we have done so demonstrating a new attack surface.
>
> We also emphasise that the novelty of this work is identifying that certifications do not just act as a defensive measure, but rather that their existence admits a heretofore undiscovered attack surface, which allows for the construction of minimal-norm attacks that are smaller than those previously seen within the literature. We would welcome any additional discussions relating to this point.

---

> ### Author Response · Authors · 2022-11-18
> **Reminder of response**
>
> We would again emphasise that the comparisons to other adversarial attack methodologies are not skewed in favour of our approach, as we are not attacking randomised model draws, but rather to the expected class output of an individual sample, after performing $N \gg 1$ draws ofrandomised smoothing. That we are considering the expected output of these models obviates the need for comparing against EOT. We also do not believe the fact that we only tested against randomised smoothing defences to be a drawback, as these are a commonly used certification mechanism, and our work is fundamentally rooted in how certification mechanisms introduce a new attack surface to models.

---

### Decision · Program_Chairs · 2023-01-20

**Decision:**

Reject

**Justification For Why Not Higher Score:**

No clear motivation for the work.

**Justification For Why Not Lower Score:**

N/A

**Metareview: Summary, Strengths And Weaknesses:**

The authors propose an attack algorithm against the SOTA certified defense, randomized smoothing. They compare their approach against several commonly used adversarial attacks and show improvements relative to those.

Strengths:
1. Comparison against SOTA attacks showing improvements.

Weaknesses:
1. Motivation: It is well-known in the adversarial robustness literature that certified defenses do not rule out adversarial examples in the general sense, only against specific threat models (commonly the l2 bounded perturbations threat model for randomized smoothing). Developing an attack here does not add much value, the key problem here is to adapt the threat model to a realistic one that capture perturbations of interest (adversarial or natural).

Hence, I recommend rejection.

**Summary Of Ac-Reviewer Meeting:**

No meeting